# Antimicrobial Efficacy of Silver Nanoparticles as Root Canal Irrigant’s: A Systematic Review

**DOI:** 10.3390/jcm10061152

**Published:** 2021-03-10

**Authors:** Shilpa Bhandi, Deepak Mehta, Mohammed Mashyakhy, Hitesh Chohan, Luca Testarelli, Jacob Thomas, Harnoor Dhillon, A. Thirumal Raj, Thodur Madapusi Balaji, Saranya Varadarajan, Shankargouda Patil

**Affiliations:** 1Department of Restorative Dental Sciences, College of Dentistry, Jazan University, Jazan 45412, Saudi Arabia; shilpa.bhandi@gmail.com (S.B.); dr.mashyakhy@gmail.com (M.M.); drhiteshchohan@yahoo.co.in (H.C.); 2Department of Preventive and Restorative Dentistry, College of Dental Medicine, University of Sharjah, Sharjah 27272, United Arab Emirates; dmehta@sharjah.ac.ae; 3Department of Oral and Maxillo Facial Sciences, Univerisity of Rome “La Sapienza”, 00161 Rome, Italy; 4Private Dental Practice, Kochi 682019, India; drjacobthomask@gmail.com; 5Private Dental Practice, Delhi 110075, India; drharnoor@gmail.com; 6Department of Oral Pathology and Microbiology, Sri Venkateswara Dental College and Hospital, Chennai 600130, India; thirumalraj666@gmail.com (A.T.R.); vsaranya87@gmail.com (S.V.); 7Department of Dentistry, Bharathirajaa Hospital, and Research Institute, Chennai 600017, India; tmbala81@gmail.com; 8Department of Maxillofacial Surgery and Diagnostic Sciences, Division of Oral Pathology, College of Dentistry, Jazan University, Jazan 45142, Saudi Arabia; dr.ravipatil@gmail.com

**Keywords:** silver nanoparticles, antimicrobials, root canal, irrigant, endodontics

## Abstract

Removal of microbes is imperative during endodontic therapy. Due to their antimicrobial property, silver nanoparticles have been used for endodontic irrigation of the root canals. The objective of the present study was to provide a qualitative analysis of the published literature assessing silver nanoparticles as root canal irrigants. A search of PubMed, SCOPUS, Web of Science, and Embase databases was done without any time restriction. Articles published in English were included. Data were extracted and the risk of bias was assessed. Of the 154 studies identified, after screening according to the inclusion criteria, five in vitro studies were included. The results indicate that silver nanoparticles have an anti-microbial effect to varying degrees depending on certain factors. Within the limitations of the present studies that have a moderate to low risk of bias, an antimicrobial effect of silver nanoparticles is observed. Silver nanoparticles have the potential to be used as endodontic irrigants, although their efficacy depends on particle size and the duration of contact which require further investigation.

## 1. Introduction

The persistence of micro-organisms in the root canal is a prominent cause of failure of endodontic treatment [1,2]. Adequate disinfection is achieved by two means: Mechanical cleaning and irrigation using antimicrobial solutions [3,4]. The paucity of time in today’s fast-paced lifestyle does not allow patients to avail themselves of multiple appointments. This creates a need for achieving complete disinfection before obturation in a single appointment. However, bacteria can persist even after cleaning, shaping, debridement, and use of antimicrobial solutions [3]. Endodontic infections are known to be polymicrobial [5]. An antimicrobial agent that acts against specific microbes such as solely gram-positive or gram-negative microbes, or aerobic microbes alone would be a poor choice in obtaining complete disinfection of the root canal. Therefore, commonly used disinfectants are sodium hypochlorite and chlorhexidine [6].

Sodium hypochlorite (NaOCl) is the preferred choice for its effective antibacterial action and the ability to dissolve tissue remnants but the presence of organic matter affects its efficacy [6]. It is generally used in concentrations ranging from 0.5% to 6% [5]. Chlorhexidine (CHX) is another widely used irrigant, equally effective as sodium hypochlorite in antimicrobial activity, but does not possess its capacity to dissolve remnants of tissue present in the canal [7,8]. An optimal root canal irrigant needs to balance between safety and effectiveness, especially in the periapical region, being effective while exerting minimal pressure in the apical region [6]. To enhance the antimicrobial efficacy of agents used in endodontics, nanoparticles and photodynamic therapy have been explored to achieve disinfection in root canals [9,10].

Nanoparticles are an area of research that has gained popularity in recent years. Nanoparticles are classified as such due to their size ranging between 1 to 100 nm. Decreased dimensions of these individual particles offer a large surface-to-volume ratio and thus can be extremely potent when they come in contact with microbes [11]. They are used as additives or can also be used as coatings for various dental materials, from implants to cement.

Silver nanoparticles (AgNPs) are a popular antimicrobial agent, though the mechanism of action is still debated [11]. Multiple modes of action of silver nanoparticles range from perforation of the cell membrane to denaturation of cellular DNA. The factors that influence the mechanism of action and which process acts during a given condition such as particle size or concentration have not been defined. Their unique properties allow them to be commonly used in dentistry. They are the NPs that have been studied the most concerning endodontics and have shown positive effects [9]. Apart from their use as an irrigant, they have been used in the form of coatings on gutta-percha to provide antimicrobial obturation and added to mineral trioxide aggregate to enhance its anti-bacterial action and improve chances of success in root canal therapy [11]. The antibacterial potential of these agents makes them interesting candidates for treating chronic infections as well [9].

Nanoparticles have been explored by other reviews that highlight their mechanism of action and use in endodontics as disinfectants [12,13,14]. These studies have explored multiple uses of silver nanoparticles as well but not as an irrigant alone. Only one among these was a systematic review that explored the efficacy of various nanoparticles [13] but did not distinguish between those used with or without adjuncts or their use as irrigants or medicaments. In the clinics, especially in cases where the treatment needs to be completed in a single appointment, placing a medication may not be an option. An irrigant with a high potential for the removal of microbes will be preferred. To understand if silver nanoparticles are effective in combating endodontic microbes, a systematic review of the literature available was conducted. The aim was to identify studies that used silver nanoparticles as irrigants in prepared root canals and to evaluate if these irrigants had an antimicrobial effect when used.

## 2. Materials and Methods

### Protocol and Registration

The international prospective register of systematic reviews (PROSPERO) database was searched for registered protocols on a similar topic. There was no registered protocol on the study of the use of silver nanoparticles and antibacterial activity. The systematic review was performed per the PRISMA (Preferred Reporting Items for Systematic reviews and Meta-Analysis) guidelines [15]. Inclusion Criteria: The PICOS format was used to include studies for review. Population (P) was teeth or sectioned tooth roots with root canals infected with any microbial organism. Intervention (I) was the use of silver nanoparticles alone in the form of a solution or a suspension as an irrigant. Comparison (C) was done with a control group which either did not have an irrigant or used conventional irrigants such as NaOCl or CHX. Outcome (O) was antibacterial activity measured using a confocal laser scanning microscope or bacterial culture. Studies (S) that assessed the association between the use of silver nanoparticles and antibacterial activity ex vivo were included. Only articles published in English were included in this systematic review. Exclusion Criteria: Narrative and systematic reviews, case/reports, letters to the editor, experimental/exploratory studies, opinion pieces, conference abstracts were excluded from the review. Articles in a language other than English were not included.

Focus Question: The research question addressed by this review was “Are silver nanoparticles, efficient antimicrobial agents, when used for root canal irrigation?” Search Strategy: A search was conducted in the PubMed, Embase, Web of Science, and SCOPUS databases without placing any time restrictions on the database up to November 2020. Only articles published in English were included in this systematic review. The free terms and medical subject headings (MeSH terms) used were endodontics, silver nanoparticles, antimicrobial, antibacterial, and irrigants.

Two reviewers independently selected studies according to the established criteria for inclusion in the systematic review. The selection process consisted of steps highlighted in Figure 1. Initial screening of titles and abstracts to identify potentially relevant studies was done. Overlapping articles were removed manually. This was followed by the examination of full texts of the selected articles to identify those which satisfied the inclusion criteria.

Only those studies which were found eligible by both reviewers were included in the review. A hand search of the references of the included studies was also done to find other potentially relevant studies.

Data obtained for analysis included the type of study, sample size, the concentration of AgNPs, alternative agents used, and the results of the study. Only those studies which used AgNP solutions/suspensions for irrigating prepared teeth were included. Assessment of Risk of Bias: A customized criteria for risk of bias was made based on previous systematic reviews [16,17]. Studies that did not report 1–3 of the items were classified as low-risk studies, those which report 4–6 items were considered as moderate bias studies, and more than 6 of those with non-reported items were considered as having a high risk of bias [17].

## 3. Results

Study selection: Identification of studies: The initial search of databases yielded a total of 154 studies. Of these studies, 122 were removed on screening the titles and the abstracts. Thirty-two articles were examined in detail. After the removal of duplicates, reviews and studies that did not fulfill the inclusion criteria were removed by both reviewers (k = 0.81). Ambiguity regarding the studies was resolved by discussion amongst the reviewers (k = 1). The remaining five articles were included in this systematic review. The studies from which data were extracted are presented in Table 1.

Study characteristics: After an examination of the results, studies that fulfilled the inclusion criteria were added to the review. A total of five studies were found [18,19,20,21,22]. These studies examined the effects of using silver nanoparticles as an irrigant in root canals when used as the only irrigant. Of the five studies included, one was from China [20]. One from India [19], one from Brazil [22], one from Brazil and France [21], and one from Iran [18]. All studies used *E. faecalis* for inoculation and four studies were on extracted human teeth (rest 4) and one was from bovine teeth [22]. The studies had varying particle sizes and used varying concentrations of AgNPs. All the studies included used Bacterial culture for detection. They examined the effect of other irrigants as well, apart from the control.

Assessment of risk of bias: The results of the assessment of the risk of bias are presented in Table 2. The criteria used were whether the groups tested human dentine and root canals for irrigation, consideration given to factors such as particle size and time of irrigation which can influence the antimicrobial activity of AgNPs and the study protocol was also taken into account. Studies included in the review consisted of 3 which were judged as having a medium risk of bias while 2 studies had a low risk of bias.

Main Findings: All the studies showed a reduction in the bacterial count with either bacterial culture or the use of live/dead staining with confocal laser scanning microscopy (CLSM). No study, however, showed a complete elimination of the micro-organisms when the AgNPs were used alone. A summary of data extracted from the studies is presented in Table 1.

## 4. Discussion

Nanotechnology is increasingly gaining a niche in the fields of medicine, especially dentistry. It has brought about the evolution of the new branch of nano dentistry which is an amalgamation of the use of nanomaterials, biotechnology, and dental nanorobotics to achieve the ideal oral health [23]. Nanotechnology has been cited as a tool to revolutionize oral healthcare. Its applications have been described in dentistry in multiple areas such as tissue engineering for dentistry, bio-nano surface technology for coatings on dental implants, targeted drug delivery systems for periodontal disease, optimizing orthodontic treatment, developing nanocomposites, and making disinfectant solutions among many others [24]. The small size of nanoparticles offers a larger surface area for a small volume of the particles. This allows them to exert action even when used in small quantities. The antimicrobial action of nanoparticles is not species-specific. It does not rely on the presence of cell wall components or specific intracellular proteins that are targeted by antibiotics. The many modes of action of silver nanoparticles make it difficult to develop resistance to their action. This is a very important aspect for endodontics today, especially with the increased use of antibiotics in healthcare.

In the field of endodontics, recurring infections and increasing resistance to antimicrobial agents possibly leading to failure of treatment are major challenges [3,25,26]. The search for an ideal disinfectant to obtain microbe-free root canals is an ongoing endeavor. Studies on failed root canal treatment have shown apical periodontitis in endodontically treated teeth generally consists of polymicrobial infections. Enterococcus faecalis (*E. faecalis*) have been isolated in many samples [27,28,29]. Other species such as Fusobacteria and Pseudomonas have been detected in later studies in higher proportions [30,31].

Different nanoparticles have been researched in the literature for their antimicrobial effects [32,33,34]. Silver is one of the oldest antimicrobials used in dentistry [35]. Silver nanoparticles have also been extensively studied for their antibacterial effects. The mechanism of action of silver nanoparticles is not known precisely. Various modes of action have been proposed. The release of silver ions which have an electrostatic attraction to sulfur proteins results in adherence and disruption of the bacterial membrane. Silver ions absorbed into the cell can disrupt respiratory enzymes generating reactive oxygen species and interrupting ATP generation. Silver ions can also cause the denaturation of DNA and ribosomes directly [11]. Apart from the release of silver ions, silver nanoparticles can also accumulate on bacterial cell walls causing denaturation of the cell wall. They can also penetrate the bacterial cell walls changing the structure of the membrane [25]. Silver nanoparticles can also interrupt signal transduction by dephosphorylation of tyrosine residues on the peptide substrates leading to cell death and apoptosis [36].

The studies included in this review utilized silver nanoparticles as irrigants in root canal disinfection. Since the aim was to evaluate the efficacy of antibacterial activity of AgNPs alone, those studies which used AgNPs as an additive to an irrigant or those which used other adjunctive materials were not included due to possible confounding. The risk of bias of the studies was done using customized criteria based on previous studies. The criteria were set to determine the strength of the study protocol, whether the conditions being tested were standardized or not. This was done by checking for sample size calculation, the extent to which simulation of the conditions, and methodology in preparing the canals as in vivo was done. We also tried to identify the character of the silver nanoparticle solution being used. This was assessed based on particle size which can affect the extent of antibacterial action. Based on these criteria, the risk of bias was assessed as low, moderate, or high.

The review followed the PICOS format to devise a research question. The screening and selection of studies were done according to the PRISMA protocol. Our research focused strictly on the use of an AgNP solution alone and did not take up any studies which modified the particles in any way. We tried to include the maximum number of studies possible but even among the studies found eligible, the characteristics of the solutions used were different. The concentration of the solution used and the particle size were different in the studies which mention it. Since the studies followed a varied methodology for the time of contact, concentration, and particle size, all of which can affect the extent of action, a meta-analysis of the data could not be done. A limitation of the review is that only ex vivo studies were found eligible.

The methodology of most studies was ex vivo but followed the protocol as close to the in vivo method of disinfection as possible. All studies except one by Wu et al. cleaned and shaped the canals before using the disinfectant solution [20]. All studies tested efficacy against *E. faecalis* which is a bacterium commonly associated with persistent infection in endodontic root canals. This was not a criterion set in the systematic review protocol. However, this limits our conclusion on the antimicrobial activity of AgNPs. Further studies on polymicrobial colonies in root canals need to be done to present a more accurate picture of the antimicrobial activity of AgNPs. Mode of evaluation differed in the studies with three studies using bacterial cell culture and two utilizing live/dead staining with a confocal laser scanning microscope (CLSM). The studies also differ regarding the particle size used the concentration of AgNPs and the time for which disinfection was done. The AgNP irrigants were delivered using a syringe in all studies though the gauge of the needle differed. This is similar to the protocol used in a clinical setup. However, the studies compared the AgNPs against various irrigants and with different contact times ranging from 1 to 30 min.

The two types of methods showed different results. Those studies which evaluated the efficacy of the solutions using CLSM found that AgNPs were not very effective in the removal of biofilms as well as the elimination of bacteria. Both studies found that the conventional irrigants, NaOCl, and CHX, show better bacterial elimination in a short contact time. The AgNPs however had a better effect compared to CHX in disruption of the biofilms at 5- and 15-min intervals in the study by Rodrigues et al. [22]. The same study also investigated the antimicrobial effect of the solutions inside the dentinal tubules. The study found that prolonged contact of the AgNP solution resulted in a greater number of live bacteria in the tubules than the biofilm. The experimental groups in both studies however had a significant improvement over the controls. Studies that evaluated antimicrobial efficacy based on bacterial cultures found a reduction in colony counts to nearly half with the use of AgNP solutions alone. In these groups, the residual nanoparticles from the solutions could have had a role to play in the prevention of the development of colonies over a longer period.

Nanoparticles have increased antimicrobial activity with a decrease in size due to an increase in the surface area which allows greater interaction between the ions and microbial organism [37]. The properties of nanoparticles are different from their corresponding bulk material, a result of the large surface to volume ratio [37]. This influences their biocompatibility and cytotoxicity as well which is much less compared to the conventional compounds [38,39,40]. It has been shown that nanoparticles have an antibacterial activity that is size-dependent. For silver nanoparticles, a size of 1–10 nm presents direct interaction with the bacteria [41]. Of the studies included in this review, those which mentioned the particle size, only one by de Almeida et al. had a particle size in this range [21]. Even here, the particle size was not uniform and had a range between 5–20 nm. Thus, the anti-bacterial action of the silver nanoparticles due to direct interaction with microbes cannot be accounted for. Two other studies had particle sizes above 10 nm and two did not mention it. It is therefore possible that the irrigants used in these studies did not utilize the complete potential of AgNPs as antimicrobial agents.

The time for which irrigation is done also appears to play an important role in the efficacy of silver nanoparticles. In studies that used a greater contact period, the antimicrobial efficacy of AgNPs was greater. Wu et al. evaluated AgNPs in the form of irrigants as well as medicaments [20]. While the irrigant solution was used for 2 min, the intracanal medicament which was a gel for AgNPs was left in situ for seven days. While the irrigation solution did not change the biofilm structure to a large extent and was not able to eliminate the bacteria, the AgNP gel exhibited structural damage to the biofilms with very few live bacteria seen using CLSM. Similarly, among the studies which evaluated the efficacy of antibacterial action using bacterial colony counts, the studies with a greater contact time showed a greater reduction in the count [18,21]. This was observed despite the aid of ultrasonic activation in the study group with a shorter contact time. 

Some of the studies presented here also examined the use of AgNPs along with the use of adjunctive procedures. Afkhami et al. [18] used a diode laser along with indocyanine green to increase the activity of AgNPs against microbes. They found the efficacy of this group to be better than the use of a diode laser or the silver nanoparticles used alone. Similarly, Kushwaha et al. [19] examined the use of silver nanoparticles along with the Nd: YAG laser. They found the efficacy of this group to be better than that of the nanoparticles alone or the use of gold nanoparticles (used with or without the Nd: YAG laser). Recent studies have found that silver nanoparticle systems with graphene oxide have potential use in reducing microbial count and disruption of the biofilm [42]. Although the use of graphene oxide can confound the effects of silver particles as it possesses antibacterial activity [43], this study presents a robust template for conducting further research on the antibacterial efficacy of nanoparticles to produce better quality studies and reduce the risk of bias.

Future studies are needed to investigate the antimicrobial properties of silver nanoparticles considering variations in the properties that influence their activity. The particle size and time of contact appear to be factors that need to be given weightage. Keeping particle size below 10 nm along with or using the irrigant solution for longer need to be researched. This will help to better evaluate the use of AgNPs in the form of root canal irrigants.

## 5. Conclusions

The data extracted from studies show that silver nanoparticles in various formulations do have an antimicrobial effect. Considering the limitations of the included studies which have a moderate to low risk of bias, it can be concluded that silver nanoparticles have an antimicrobial effect when used to irrigate root canals. They have the potential to be used as endodontic irrigants. The efficacy of these nanoparticles however depends on factors such as particle size and the duration of contact which require further investigation to better understand their usage as an endodontic irrigant.

## Figures and Tables

**Figure 1 jcm-10-01152-f001:**
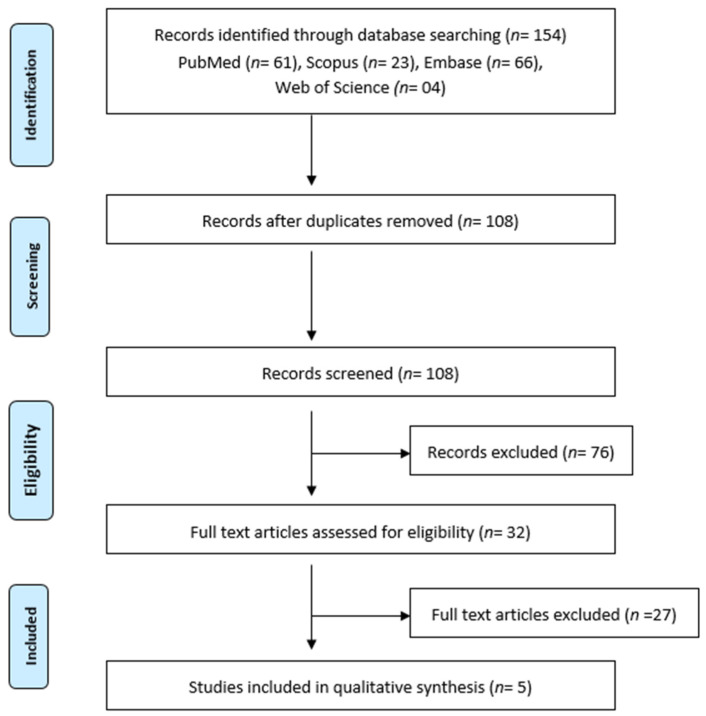
PRISMA Flowchart depicting the workflow of the systematic review.

**Table 1 jcm-10-01152-t001:** Summary of Data extracted from studies.

S. No.	Author/Year/Country	Type of Study	Specimens Used	Bacterial Inoculation Was Done with	Experimental Group/Group of Interest	Control Group	Other Groups Studied	Method of Detection	Results	Inference	Conclusion
1	Afkhami/2017/Iran [18]	in vitro	Extracted human teeth	*E. faecalis*	AgNP suspension (100 ppm, particle size 30 nm) for 5 min	2.5% NaOCl for 5 min	Diode laser, Indocyanine green with a diode laser, AgNPs with a diode laser, and Indocyanine green	Bacterial culture	94.42% reduction in colony count	100 ppm solution of AgNp and 2.5% NaOCl showed similar antimicrobial efficacy	AgNPs are effective antimicrobials for root canal irrigation.
2	Kushwaha/2018/India [13]	in vitro	Extracted human teeth	*E. faecalis*	AgNP suspension (100 ppm, particle size 20 nm) for 3 min	2% CHX	Gold NPs, AgNP with Nd: YAG laser, GoldNPs with Nd: YAG laser	Bacterial culture	Significant reduction in colony-forming units	AgNPs with Nd: YAG irradiation is a better irrigant than the other groups	AgNPs alone are effective but combined with Nd: YAG has a better effect than all other groups
3	de Almeida/2018/Brazil, France [15]	in vitro	Extracted human teeth	*E. faecalis*	1% AgNP solution (particle size 5–20 nm) activated ultrasonically for 1 min	0.85% saline ultrasonically activated for 1 min	2% CHX, 1% NaOCl, 5% NaOCl, 26% ZnNP (all solutions ultrasonically activated for 1 min)	Bacterial culture	57.28% reduction in colony-forming units	AgNPs significantly decrease the number of viable CFUs in the biofilm	1% AgNP was able to reduce *E. faecalis* biofilm in root canals similar to conventional irrigants
4	Rodrigues/2018/Brazil [16]	in vitro	Bovine central incisors	*E. faecalis*	1 mL AgNP solution (particle size 94 ppm) for 5, 15, 30 min	1 mL saline solution with contact times of 5, 15, and 30 min)	1 mL each of 2% CHX and 2.5% NaOCl with the same contact time as the control	Staining with live/dead technique and evaluation with a confocal laser scanning microscope	Ag NP used for 5 min had bacteria present in the biofilm while the 30 min group had bacteria in the dentinal tubules	AgNP solution was significantly less effective than NaOCl	AgNPs were not effective in dissolving the biofilm or eliminating the bacteria
5	Wu/2014/China [14]	in vitro	Vertically sectioned extracted human teeth	*E. faecalis*	6 mL of 0.1% AgNP solution for 2 min	No irrigation	2% NaOCl, sterile saline	Staining with live/dead technique and evaluation with a confocal laser scanning microscope	Most biofilms were intact after irrigation with the 0.1% AgNP solution	A 2 min irrigation with 0.1% Ag NP solution provides a limited antibacterial effect	0.1% AgNP solution does not significantly kill bacteria or alter the biofilm structure

**Table 2 jcm-10-01152-t002:** Assessment of risk of bias.

S. No.	Author/Year/Country	Were Human Teeth Used as Specimens?	Was the Sample Size Statistically Calculated?	Was Bacterial Inoculation Verified?	Was Particle Size Mentioned?	Was a Control Group Present?	Were the Teeth Cleaned and Shaped before Irrigation?	Was the Irrigation Time the Same for Experimental and Control Groups?	Was the Observer/Evaluator Blind to the Groups?	Was There Any Conflict of Interest?	Risk of Bias
1	Afkhami/2016/Iran [18]	Yes	Not reported	Yes	Yes	Yes	Yes	Yes	Not reported	None	Low
2	Kushwaha/2018/India [13]	Yes	Not reported	No	Yes	Yes	Yes	Yes	Not reported	Not reported	Medium
3	de Almeida/2018/Brazil, France [15]	Yes	Not reported	Yes	Yes	Yes	Yes	Yes	Not reported	None	Low
4	Rodrigues/2018/Brazil [16]	No	Not reported	Yes	Not mentioned	Yes	Yes	Yes	Not reported	None	Medium
5	Wu/2014/China [14]	Yes	Not reported	Yes	Not mentioned	Yes	Not reported	Yes	Not reported	None	Medium

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
