# Peer review of "Antimicrobial Efficacy of Silver Nanoparticles as Root Canal Irrigant’s: A Systematic Review"

_jcm, 2021, doi:10.3390/jcm10061152_

Round 1

Reviewer 1 Report

This research is under the scope of this journal; the topic is relevant for readers and this research deals with potentially significant knowledge to the field and open a new way for future studies. The aim of this paper is quite interesting.

However, there are some concerns about the present manuscript:

(Introduction)

  • What is the importance of this review for the clinical? You do not think this study is included in the others already done? Which results are comparable?
  • What this study has new?
  • To support the sentence about mechanical and chemical technique. https://doi.org/10.1007/s10266-018-0401-2
  • The nanoparticles combine with other photosensitiser https://doi.org/10.3390/jfb10040044
  • Table 1 - ml change for mL.

(References)

  • In the manuscript, the insertion was not standardized. Please, insert the reference in the text before the end-point.
  • Table 1 and 2 – also add the references
  • References are not standardized:

The titles of references have a different format, the title of the article is written in capital letters at the beginning of words, others only in lower case. Also, the standardized format of presentation in the journal's name. Because names have written in a different format, one is not abbreviated, others are not.

Author Response

We thank the reviewers for their interest and feedback on the article.

The changes that were suggested by the reviewers have been incorporated in the article. We have also tried to answer the individual questions posed to the best of our knowledge.

Reviewer 1:

  • What is the importance of this review for the clinical? You do not think this study is included in the others already done? Which results are comparable?

A clinician is likely to use endodontic irrigants more with growing popularity of single visit endodontic treatments. This review evaluates the studies that used silver nanoparticles specifically, as an endodontic irrigant. While former reviews have explored their antimicrobial properties, none has focused exclusively on silver nanoparticles alone as an irrigating solution i.e. they have been used as with adjuncts. We evaluate whether silver nanoparticles exclusively can be used as irrigants.

Changes were made in the introduction to highlight the same. (line 71-78) 

  • What this study has new?

This study focuses exclusively on silver nanoparticles and irrigation. We found that:

  • The size of nanoparticles which can affect their antimicrobial property and also the contact time was not considered in the studies. (We highlight in Discussion, lines 245 - 269 and conclusion)
  • The method of evaluation affects the inference drawn by the authors. (Discussion, lines 231 – 244)
  • To support the sentence about mechanical and chemical technique. https://doi.org/10.1007/s10266-018-0401-2
  • Added as reference no. 4
  • The nanoparticles combine with other photosensitiser https://doi.org/10.3390/jfb10040044
  • Added as reference no. 10
  • Table 1 - ml change for mL.
  • Changes made in the table accordingly

(References)

  • In the manuscript, the insertion was not standardized. Please, insert the reference in the text before the end-point.
  • Table 1 and 2 – also add the references
  • References are not standardized:

The titles of references have a different format, the title of the article is written in capital letters at the beginning of words, others only in lower case. Also, the standardized format of presentation in the journal's name. Because names have written in a different format, one is not abbreviated, others are not.

  • Changes were made in the document according to the above suggestions to the document.

Reviewer 2 Report

The authors performed a systematic literature review without meta-analysis of the data on the efficacy of silver nanoparticles as a root canal irrigant. The systematic review could find interest in the endodontic field

There are some suggestions that can certainly help to improve the systematic review

  • Report for each keyword used, the number of articles (records) identified in each single database (PUB MED, EMBASE, Scopus, web of science)
  • Report the search period on the databases and the last search date.
  • It would be advisable to redo the flow chart using the PRISMA template
  • Report the overlaps removal method (manual or by software for example: endnote)
  • Report the degree of agreement (k agreement) between the 2 reviewers in the choice of articles
  • Report how the disagreements between the 2 reviewers were resolved:
  • Add bibbliographic reference to the following statement: An antimicrobial agent that acts against specific microbes such as solely gram-positive or gram-negative microbes, or aerobic microbes alone would be a poor choice in obtaining complete disinfection of the root canal. Therefore, commonly used disinfectants are sodium hypochlorite and chlorhexidine
  • I wouldn't be so sure about this statement:, (Chlorhexdine (CHX) is another widely used irrigant, equally effective as sodium hypochlorite in antimicrobial activity, but does not possess its capacity to dissolve remnants of tissue present in the canal).  it with more recent bibliographical references.
  • Bibliographic references must be inserted before the point, not after and not at the beginning of a sentence: [6] and [9]
  • What innovations brought about by the reviews already present in the scientific literature? [1-4]

  1. Shrestha, A .; Kishen, A. Antibacterial Nanoparticles in Endodontics: A Review. J Endod 2016, 42, 1417-1426, doi: 10.1016 / j.joen.2016.05.021.
  2. Shashirekha, G .; Jena, A .; Mohapatra, S. Nanotechnology in Dentistry: Clinical Applications, Benefits, and Hazards. Compendium of continuing education in dentistry (Jamesburg, N.J.: 1995) 2017, 38, e1-e4.
  3. Samiei, M .; Farjami, A .; Dizaj, S.M .; Lotfipour, F. Nanoparticles for antimicrobial purposes in Endodontics: A systematic review of in vitro studies. Materials science & engineering. C, Materials for biological applications 2016, 58, 1269-1278, doi: 10.1016 / j.msec.2015.08.070.
  4. Raura, N .; Garg, A .; Arora, A .; Roma, M. Nanoparticle technology and its implications in endodontics: a review. Biomaterials research 2020, 24, 21, doi: 10.1186 / s40824-020-00198-z.

Author Response

We thank the reviewers for their interest and feedback on the article.

The changes that were suggested by the reviewers have been incorporated in the article. We have also tried to answer the individual questions posed to the best of our knowledge.

Reviewer 3 Report

This systematic review aims to evaluate the Antimicrobial efficacy of silver nanoparticles as root canal irrigants.

General overview: It is a well-designed work, clearly exposed and understandable to readers. Provides clear and sufficiently detailed information.

Discussion

P 6 line 169-170

Studies on failed root canal treatment have shown that the predominant species of bacte169 ria found in treated canals with apical periodontitis generally consist of enterococci and 170 Enterococcus faecalis (E.faecalis) have been isolated in most of the samples.

It will be added that other microorganisms have been find in greater proportions that E faecalis in failed root canal treatment. You can see these references for example.

Schirrmeister JF, Liebenow AL, Pelz K, Wittmer A, Serr A, Hellwig E, et al. New Bacterial Compositions in Root-filled Teeth with Periradicular Lesions. J Endod. 2009;35:169-74. 13.

Siqueira JF, Antunes HS, Rôç IN, Rachid CTCC, Alves FRF. Microbiome in the apical root canal system of teeth with post-treatment apical periodontitis. PLoS One. 2016;11:1-14.

P 6 line 187-188

review these phrases

Those studies which used AgNPs as an additive to an irrigant were not in187 cluded. Those studies where AgNPs were modified or used in combination with another 188 agent were not included due to possible confounding.

Author Response

We thank the reviewers for their interest and feedback on the article.

The changes that were suggested by the reviewers have been incorporated in the article. We have also tried to answer the individual questions posed to the best of our knowledge.

Reviewer 3:

Discussion

P 6 line 169-170

Studies on failed root canal treatment have shown that the predominant species of bacte169 ria found in treated canals with apical periodontitis generally consist of enterococci and 170 Enterococcus faecalis (E.faecalis) have been isolated in most of the samples.

It will be added that other microorganisms have been find in greater proportions that E faecalis in failed root canal treatment. You can see these references for example.

Schirrmeister JF, Liebenow AL, Pelz K, Wittmer A, Serr A, Hellwig E, et al. New Bacterial Compositions in Root-filled Teeth with Periradicular Lesions. J Endod. 2009;35:169-74. 13.

Siqueira JF, Antunes HS, Rôç IN, Rachid CTCC, Alves FRF. Microbiome in the apical root canal system of teeth with post-treatment apical periodontitis. PLoS One. 2016;11:1-14.

  • The studies mentioned above were added to the discussion in Lines 179 - 180

P 6 line 187-188

review these phrases

Those studies which used AgNPs as an additive to an irrigant were not in187 cluded. Those studies where AgNPs were modified or used in combination with another 188 agent were not included due to possible confounding

  • The sentence was rewritten, Lines 195 -197

Reviewer 4 Report

Τhank you for submitting your manuscript in J Clin Med. This an interesting literature review. Taking into account that literature is scarce and the selected papers are low in numbers and quantity, I would suggest the addition of the paper attached in the word file, as well as search for more papers from the research team of University of Toronto Prof Anil Kishen. 

The review could be even more interesting if you included more nanoparticles implemented for endodontic disinfection.

Please ensure you follow my instructions to improve quality of paper by adding the paper and re searching the literature

Author Response

Dear reviewer,

We thank you for your time and valuable feedback on the manuscript.

The studies included in the review have a low to moderate risk of bias with the majority having a medium risk of bias. We have stated the same in the result lines 151 – 152.

We searched the extensive work by Dr. Kishen and his team. We have already included one of his articles in the review[1]. He has also written an extensive and educational review on nanoparticles[2] which was referred to in the introduction. In addition, his work has also focused on chitosan and zinc nanoparticles. However, as we tried to establish the efficacy of silver nanoparticles only, we were unable to include the same in the review.

We read the paper suggested for inclusion[3]. This study presents a robust methodology for assessment of antibacterial action of an irrigant solution. In this study, the authors have used a silver nanoparticle graphene oxide system for irrigation of root canals and compared it to other systems. The use of graphene oxide can introduce possible confounding[4]. We, therefore, were unable to introduce the study into the review. However, we have included a recommendation in the discussion (lines 279 – 285) for this study’s protocol serving as a template for future research.

  1. Wu, D.; Fan, W.; Kishen, A.; Gutmann, J.L.; Fan, B. Evaluation of the antibacterial efficacy of silver nanoparticles against {Enterococcus} faecalis biofilm. J. Endod. 2014, 40, 285–290, doi:10.1016/j.joen.2013.08.022.
  2. Shrestha, A.; Kishen, A. Antibacterial Nanoparticles in Endodontics: A Review. J. Endod. 2016, 42, 1417–1426.
  3. Ioannidis, K.; Niazi, S.; Mylonas, P.; Mannocci, F.; Deb, S. The synthesis of nano silver-graphene oxide system and its efficacy against endodontic biofilms using a novel tooth model. Dent. Mater. 2019, 35, 1614–1629, doi:10.1016/j.dental.2019.08.105.
  4. Kumar, P.; Huo, P.; Zhang, R.; Liu, B. Antibacterial properties of graphene-based nanomaterials. Nanomaterials 2019, 9, 737.

Round 2

Reviewer 1 Report

This research is under the scope of this journal; the topic is interesting for readers and this research deals with potentially significant knowledge to the field and an open new way for future studies. 
The authors improved the quality of the manuscript after the reviewer's indications.

Author Response

Dear Reviewer, 
Thank you for your kind comments.
Best Regards.

Reviewer 2 Report

The authors made all required changes.

Author Response

(The authors gave the same response as above.)
